# Investigation of Nonlinear Optical Modulation Characteristics of MXene VCrC for Pulsed Lasers

**DOI:** 10.3390/molecules27030759

**Published:** 2022-01-24

**Authors:** Yao Meng, Yizhou Liu, Tao Li, Tianli Feng, Jiacheng Huang, Zheng Ni, Wenchao Qiao

**Affiliations:** 1School of Information Science and Engineering, Shandong University, Qingdao 266237, China; mengyaomyy@163.com (Y.M.); yizhouliu@sdu.edu.cn (Y.L.); litao@sdu.edu.cn (T.L.); tlfeng@sdu.edu.cn (T.F.); hhhhjc0106@163.com (J.H.); 2Shandong Provincial Key Laboratory of Laser Technology and Application, Shandong University, Qingdao 266237, China; 3China Key Laboratory of Laser & Infrared System (Ministry of Education), Shandong University, Qingdao 266237, China; 4CHN Energy Shouguang Company, Weifang 262714, China; 16090255@chnenergy.com.cn

**Keywords:** MXenes, solid-state laser, Q-switching

## Abstract

We report the surface morphology and the nonlinear absorption characteristics of MXene VCrC nanosheets prepared by the liquid-phase exfoliation method. The self-made MXene VCrC was applied as a saturable absorber in the Tm:YAP laser experiments, performing excellent Q-switching optical modulation characteristics in infrared range. With this absorber, a stable passively Q-switched 2 μm laser was achieved. Under an incident pump power of 3.52 W, a maximum output power of 280 mW was obtained with a T = 3% output coupler at a repetition frequency of 49 kHz. The corresponding pulse energy and peak power were 5.7 μJ and 6.6 W, respectively. The shortest pulse duration was 658 ns at the repetition rate of 63 kHz with a T = 1% output coupler.

## 1. Introduction

Near-infrared (NIR) pulsed lasers have attracted more and more attention in various fields, such as laser surgery, material processing, environmental monitoring, and free-space communication [1,2,3]. Q-switching technology has been demonstrated as a direct method in pulsed laser generation. Compared with active Q-switching technology, passive Q-switching (PQS) technology utilizing saturable absorbers (SAs) has advantages such as no additional drive, low cost and compactness [4,5,6]. In the past years, many novel nanomaterial SAs, such as graphene, transition metal dichalcogenides (TMDs), topological insulators, and black phosphorus [7,8,9,10], have come to our attention due to their excellent optical properties, such as the wide absorption spectrum range and significant saturable absorption capacity [11,12].

MXenes are a novel group of the two-dimension (2D) nanomaterials with Van der Waals layered structure, quantum Hall effect, excellent electrical and thermal conductivity and good stability [13,14]. The general formula of MXenes can be expressed as M_n+1_X_n_T_x_, wherein M refers to a transition metal element, X refers to element C or N, T represents the functional groups (O, F, and OH), and *n* = 1, 2, 3 [15]. MXenes have been widely used in the fields of energy storage, photo-electrochemical catalysis, photonics, and sensors, proved to be suitable for being utilized as the optical modulation devices [16,17]. Up to now, MXenes have been applied as the saturable absorbers to realize multiple Q-switched pulsed lasers. Huang et al. built a Q-switched Nd:YAG laser with the Ti_2_CT_X_ MXene with the minimum pulse width of 163 ns at 1.06 μm [18]. It was reported that a Q-switched Tm, Gd:CaF2 laser at 1.93 μm band was demonstrated by Zu et al., applying the Ti_3_C_2_T_X_ MXene [19]. Niu et al. obtained a Tm:YAP Q-switched laser with the Nb_2_CT_X_ MXene SA, generating 1.96 μs pulse width at 1.94 μm [20].

Further investigations on novel 2D materials in developing the Q-switched pulsed lasers with high operating performance are still necessary for advancing further applications. The ordered double transition metals carbides M’M”C have been recently found as a new member of the MXenes family. The combination of both metals and mixed carbon-nitrides might potentially expand the range of accessible properties and applications greatly [21]. Recently, a new layered MXene VCrC with excellent metallic properties with near-zero energy band gap has aroused the interests of researchers. To our knowledge, there are no reports of utilizing the MXene VCrC as the saturable absorber in the passively Q-switched 2 μm laser. Further investigations on its surface morphology and the nonlinear absorption characteristics turn out to be urgent to satisfy requirements in developing novel passively Q-switched lasers.

In this paper, we successfully synthesized the MXene VCrC nanosheets SA and then studied the surface morphology and the nonlinear saturable absorption characteristics. Stable passively Q-switched Tm:YAP pulsed laser was realized with the homemade SA at 2 μm for the first time. The maximum output power was 280 mW under an incident pump power of 3.52 W with a T = 3% output coupler. Meanwhile, the pulse duration was measured to be 865.3 ns with a repetition frequency of 49 kHz, corresponding to pulse energy and peak power of 5.7 μJ and 6.6 W, respectively. The obtained shortest pulse duration was 658 ns at a repetition rate of 63 kHz with a T = 1% output coupler of which the temporal profiles of the pulse train were presented at the end of the paper.

## 2. Preparation and Characterization of MXene VCrC SA

### 2.1. Preparation of MXene VCrC SA

In this experiment, the MXene VCrC nanosheets SA was prepared by the liquid-phase exfoliation (LPE) method. Firstly, the VCrC MXene powder (5 mg) and absolute ethanol solution (5 mL) were intensively mixed in a centrifuge tube and ultrasonically shaken for 2 h. After being centrifuged at a rotating of 5000 rpm for 5 min, the upper 1/3 supernatant solution was dropped onto a quartz window with a size of (20 × 20 × 0.5) mm^3^. Then, the quartz window was delivered on the spin coater and spin-coated at a speed of 400 rpm. Finally, the MXene VCrC SA was fabricated successfully after being air-dried at RT.

### 2.2. Characterization and Analysis

Firstly, the surface morphology of the MXene VCrC nanosheets was characterized utilizing the scanning electron microscope (SEM), shown in Figure 1a. The visible layered structure could be observed with different spatial resolution, indicating the successful etching effect. According to the results of the X-ray diffraction (XRD) pattern exhibited in Figure 1b, three sharp typical diffraction peaks indicated the good crystallization of MXene VCrC. The height variation was characterized by the atomic force microscopy (AFM, HORIBA Smart SPM, Horiba, Kyoto, Japan), shown in Figure 1c,d. The thickness distribution of the VCrC MXene was distributed in the range of 70–150 nm, showing the sample had been successfully exfoliated into multilayered films with a sub-micron scale.

### 2.3. Optical Absorption Properties

The linear transmission spectrum of the MXene VCrC nanosheets in the wavelength range from 500–2500 nm was further characterized by a UV-vis-NIR spectrophotometer (SHIMADU UV3600, Shimadzu, Kyoto, Japan). As shown in Figure 2a, the MXene VCrC nanosheets exhibited certain optical absorption at 2 μm region. The plotted fluctuation contents of the transmission curve at 650 and 1650 nm were caused by tuning the reference lamp.

The nonlinear optical absorption properties of the MXene VCrC nanosheets were further investigated by the open-aperture (OA) Z-scan technology, as shown in Figure 2b. The Z-scan measurement was performed with the ~2 µm pulsed laser source of 800 kHz repetition rate and 100 ns pulse width. The incident pulse fluence on the sample could be adjusted by shifting the position of sample on the Z axis. The experimental data of Z-scan can be fitted with the equation [22]:(1)T=∑m=0∞[−q0(z,0)]m(m+1)1.5,m∈N  q0(z,0)=βeffLeffI0(1+z2z02),where $L_{eff}=\left(1-e^{L\alpha_{0}}\right)/\alpha_{0}$ is the effective length, *α*_0_ is the linear absorption coefficient, *L* is the sample length, *I*_0_ represents the on-axis peak intensity, and $\beta_{eff}$ represents the effective nonlinear absorption indices. As shown in Figure 2c, the normalized transmission curve indicated a symmetrical peak relative to *Z*_0_, evidencing that MXene VCrC nanosheets could be applied as saturable absorbers based on the considerable saturation absorption capability at 2 µm.

To further analyze the saturable absorption characteristics in detail, the nonlinear transmission of VCrC versus incident pump intensity was measured as exhibited in Figure 2d. The modulation depth, the nonlinear saturation loss, and the saturation intensity were obtained by fitting experimental data with the following transmission formula [23]:(2)$$T=1-\Delta Texp\left(-\frac{I}{I_{s}}\right)-T_{ns},$$

The fitting curve indicated that the saturation fluence *I_s_*, the modulation depth Δ*T*, and the non-saturable loss *T_ns_* of the MXene VCrC nanosheets were 2.6 MW/cm^2^, 7.0%, and 5.4%, respectively. Irradiated on the pulsed laser at 2 μm wavelength region with the pulse widths of 100 ns and spot radius of ~100 μm, the sample was not damaged, so the laser-induced damage threshold should be estimated above 13 MW/cm^2^.These results revealed that the as-prepared VCrC sample could be utilized as the potential saturable absorbers for Q-switched lasers. In addition, the large modulation depth would lead to the short pulse width.

## 3. Laser Experiment and Results

In order to further confirm the nonlinear absorption properties of the prepared MXene VCrC at 2 µm region, a passively Tm:YAP Q-switched laser was obtained with a 20-mm-long plane-parallel cavity shown in Figure 3a. The pump source was a 794 nm fiber-coupled diode laser with a core diameter of 400 μm and numerical aperture of 0.22. Through the optical refocus module (1:1) composed of two plano-convex lenses with focal lengths of 50 mm, the pump beam was focused onto the surface of the gain medium with a beam radius of 200 μm. The gain medium was wrapped with indium foil and mounted on a water-cooled copper heat-sink maintained at 15 °C to significantly remove the thermal load during the laser operation. The input plane coupling mirror M1 was antireflective coated at 790 nm and highly reflective coated at 2 µm. The OC M2 was also a flat mirror with the transmittance at 2 μm of 1%, 3%, and 5%, respectively. A filter was put behind the mirror M2 to block the pump light. We applied a power meter to measure the output power and made use of a digital phosphor oscilloscope combined with a photodetector to record the pulse temporal behavior.

Firstly, we investigated the CW output power of laser performances without the VCrC SA, which are represented in Figure 3b. The lasing thresholds of the pump powers were 1.5, 1.8, and 1.7 W with T = 1%, 3%, and 5% OCs, respectively. The CW output power increased linearly with the incident pump power and the corresponding slope efficiencies were 26.9%, 29.8%, and 32.3%, respectively. With the VCrC SA inserted, the PQS lasing threshold pump powers were 1.80, 2.18, and 2.36 W for above mentioned OCs. The PQS output power was also linearly increased with the pump power as shown in Figure 3c. We obtained a maximum average output power of 280 mW at the incident pump power of 3.52 W with the T = 3% OC, corresponding to a slope efficiency of 19.4%. Both the CW and PQS lasing spectra were shown in Figure 3d. The CW output wavelength was centered at 1994.7 nm while the PQS output central wavelength blueshifted to 1980.0 nm, which may be attributed by the high insertion loss in the resonator that influenced the mode competition of CW operation and resulted in rebuilding new mode competition under the PQS operation. 

Figure 4 summarizes the variation trends of the pulse duration, repetition rate, single-pulse energy and peak power versus the incident pump powers. The pulse duration declined along with the increase of the incident pump power while the pulse repetition rates increased. Under an incident pump power of 3.52 W, the shortest pulse durations at T = 1%, 3%, and 5% were 658, 865.3, and 883 ns, respectively. Within the pump power range, the repetition rates increased from 26 to 63 kHz, 30 to 49 kHz, 19 to 53 kHz for T = 1%, 3%, and 5% OCs, respectively. Figure 4c,d shows the variation of single-pulse energy and peak power against the incident pump power. The obtained maximum single-pulse energies were 1.9, 5.7, and 3.6 μJ, corresponding to the peak powers of 2.9, 6.6, and 4.1 W. From Figure 4e, the temporal pulse train was obtained at the repetition rate of 63 kHz at T = 1% and single-pulse shape with the shortest pulse duration of 658 ns indicated the stable Q-switching performance of the VCrC SA. 

Table 1 summarizes the Q-switched laser performance of the MXene VCrC and other 2D nanomaterials, including black phosphorus, MoS_2_, Ti_3_C_2_T_X_, etc. We can see that the VCrC sample possessed good optical modulation characteristics and proved to be a promising saturable absorber candidate for pulsed laser at the NIR region.

## 4. Conclusions

In conclusion, the layered MXene VCrC nanosheets were successfully fabricated by LPE method. The corresponding surface morphology and nonlinear absorption characteristics were carefully investigated. Employing the prepared MXene VCrC SA, a stable passively Q-switching Tm:YAP laser at 2 μm was realized. To our best knowledge, this is the first demonstration of MXene VCrC SA utilized in realizing pulsed lasers. Under an incident pump power of 3.52 W, we obtained a maximum output power of 280 mW with a T = 3% output coupler at repetition frequency of 49 kHz and a pulse duration of 865.3 ns corresponding to the pulse energy and the peak power of 5.7 μJ and 6.6 W, respectively. The shortest pulse duration was measured to be 658 ns at a repetition rate of 63 kHz with a T = 1% output coupler. The excellent experimental results indicated the promising potential of MXene VCrC SA in generating optical modulation in solid-state lasers operating at the NIR region.

## Figures and Tables

**Figure 1 molecules-27-00759-f001:**
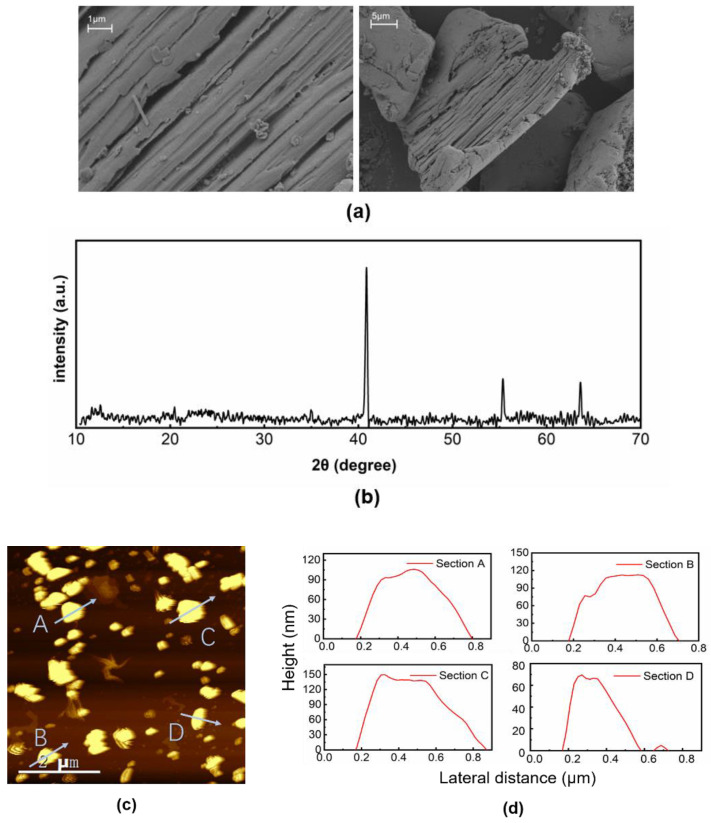
Systematic characterization of the prepared MXene VCrC: (**a**) SEM image, (**b**) XRD pattern, (**c**) AFM image, and (**d**) height profiles.

**Figure 2 molecules-27-00759-f002:**
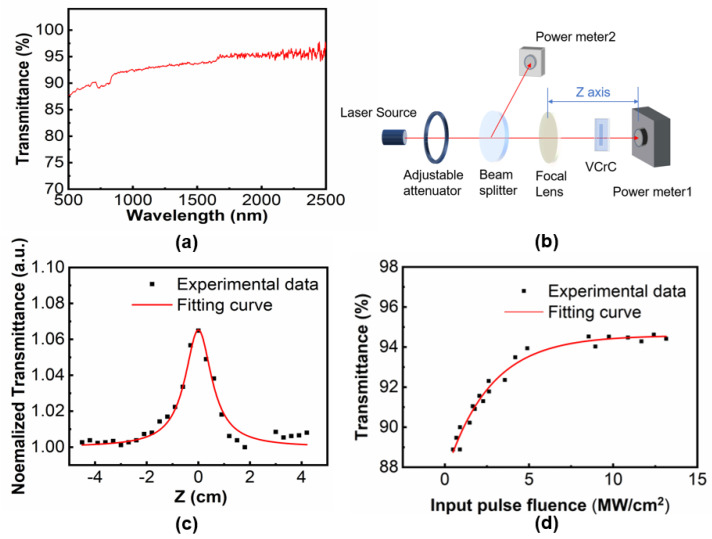
(**a**) Linear optical transmittance spectrum of the MXene VCrC SA. (**b**) Schematic diagram of the open-aperture (OA) Z-scan. (**c**) OA Z-scan curve and (**d**) nonlinear transmittance curve.

**Figure 3 molecules-27-00759-f003:**
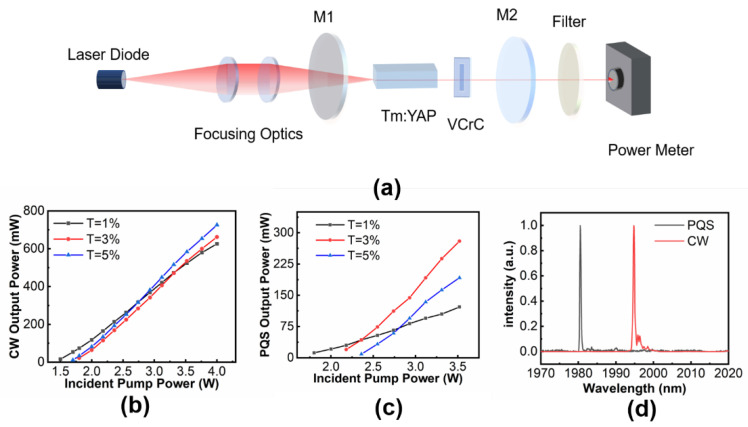
(**a**) Schematic diagram of the MXene VCrC Q-switched Tm:YAP laser experiment. (**b**) Output powers versus incident pump powers for Tm:YAP lasers in CW regime and (**c**) PQS regime. (**d**) The output spectra for the Tm:YAP lasers.

**Figure 4 molecules-27-00759-f004:**
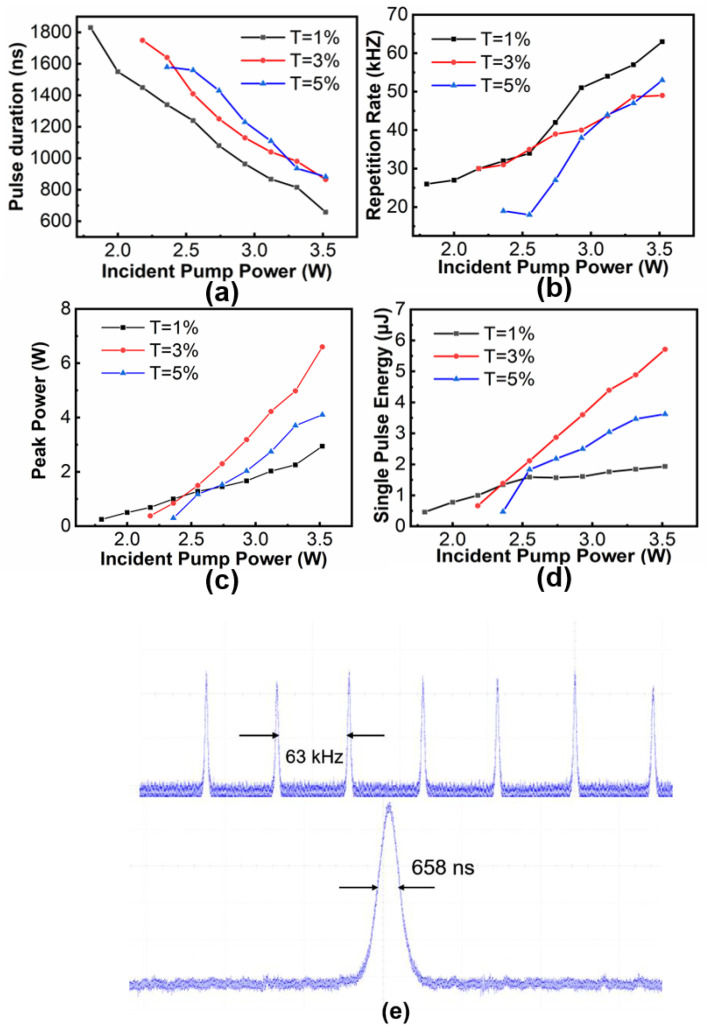
(**a**) Pulse duration; (**b**) pulse repetition rate; (**c**) single-pulse energy and (**d**) peak power versus the incident pump powers from the PQS laser. (**e**) Typical pulse train and temporal pulse profile.

**Table 1 molecules-27-00759-t001:** Performance comparison of PQS lasers with different SAs at 2 μm region.

SA	Output Power (mW)	Pulse Duration (ns)	Pulse Repetition Rate (kHz)	Single Pulse Energy (μJ)	Peak Power (W)	Reference
MoS_2_	410	458.8	83.1	4.93	10.7	[10]
WS_2_	668	528.4	87.7	7.62	14.4	[10]
Mg-MOF-74	660	313	117	5.6	18.0	[24]
BP	151	1780	19.3	7.84	4.4	[25]
Ag-NR	385	3100	9.3	41.4	13.3	[26]
graphene	310	285	190	1.6	6	[27]
Ti_3_C_2_T_X_	208	2390	19.6	10.61	4.44	[19]
Nb_2_CT_X_	623	1960	80	7.78	3.97	[20]
VCrC	280	865.3	49	5.7	6.6	this work

## Data Availability

Data are contained within this article.

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
