# Peer review of "Investigation of Nonlinear Optical Modulation Characteristics of MXene VCrC for Pulsed Lasers"

_molecules, 2022, doi:10.3390/molecules27030759_

Round 1

Reviewer 1 Report

The authors report on the preparation, optical and Q-switching properties of the new MXene VCrC. 
All in all, everything is fine, hence I would recommend to publish. However, I missed a short discussion on whether the reported laser properties are consistent with the measured optical properties of the saturable absorber. 

Since I am no expert in the chemistry / preparation I will comment on some details concerning the optical part of the paper.

1) Abstract:
"meanwhile" is not relevant for the abstract. Just give the relevant data. Do the pump power (3.5 W), average power (280 mW) and repetition rate 49 kHz) of the "meanwhile" experiments also apply to these "meanwhile" results?

2) Optical characterization
Looking at the microscopic images (Fig. 1), I wonder whether there is small and large angle scattering, whether this would have been measured with the transmittance setup (which depends on the numerical aperture of the setup). Have you measured whether there is a degradation of the beam quality after passing the sample?
What was the beam size on the VCrC at focus? Have you seen any changes when moving the sample laterally?
Comparing Fig. 2d and 2a I would conclude that the saturation measurements have been done around 532 nm. Are there any saturation data for 2 µm, where the Q-switching was done. According to Fig. 2a, there is a difference of the transmittance losses of a factor of 2 between 0.5 and 2 µm.
What are the reflectance losses? Are they also intensity dependent? 
Later on in the laser experiments, if the sample is normal to the resonator axis, the reflectance part of the transmittance losses may not be relevant for the resonator losses.
Could you please explain eq. (1)? I understand that q0 is based on the caustic of the laser beam. I do not understand why the calculation of Leff is based on the unsaturated absorption coefficient. Is the sum to take into account possible multi-passes inside the sample. If yes, wouldn't that mean that the reflectivty of the surface had to be taken into account?

3) Laser experiments
Two important data are missing here: the doping level and length of the Tm:YAP.
The pump radius is approximately constant over the length of the YAP? Looking at Fig. 3a one would conclude this is not.
The mode size, both in the YAP and the VCrC is approx. equal the pump spot size?
Do you have any information on the beam quality (M²)?
I would very much like to see a short discussion whether the laser parameter (threshold and slope) are roughly in agreement with the saturation parameters measured above (in case they apply to 2 µm):
The cw threshold is only slightly depending on the OC. Hence this should be mainly attributed to the quasi-3-level threshold (thermal population of the lower laser level). Is this consistent with the theoretically expected threshold? To calculate this, the doping level and the laser mode volume needs to be known.
The increase of the threshold and the decrease of the slope, is this consistent with the more or less saturated losses of the absorber? 
Assuming the 3 W peak power for OC = 3 %, I calculated a peak intensity of 300 kW/cm², which would still be well below the saturation intensity. Is that correct? 
Table 1: For a fair comparison, one should also know the pump power. Perhaps one could include also the saturation parameters of the materials listed.

Author Response

We would like to thank the reviewer for the thoughtful review of our manuscript. Below is our point-by-point response to the reviewer’s comments.

Reviewer 2 Report

Report on "Investigation of nonlinear optical modulation characteristics of 2
MXene VCrC for pulsed lasers".

In this contribution authors studies novel saturable absorber for 2 um solid-state laser. In my opinion, this paper can be interesting for the laser community.

Please find below my comments that should improve the quality of this paper:

  1. Please provide the damage threshold for this new saturable absorber.
  2. Please provide more information about the experiment for example the accurate focal length of the lens used for coupling light to crystal.

Author Response

Thank you for the constructive comments and advice. Below is our point-by-point response to the reviewer’s comments.
